

# Beyond width and density: stable carbon and oxygen isotopes in cork-rings provide insights of physiological responses to water stress in *Quercus suber* L

Augusta Costa[1,2], Paolo Cherubini[3,4], José Graça[5], Heinrich Spiecker[6], Inês Barbosa[2] and Cristina Máguas[7]

[1] Instituto Nacional de Investigação Agrária e Veterinária, I.P., Oeiras, Portugal
[2] Center for Environmental and Sustainability Research, NOVA University of Lisbon, Caparica, Portugal, Caparica, Portugal
[3] Swiss Federal Research Institute WSL, Birmensdorf, Switzerland
[4] Department of Forest and Nature Conservation, Faculty of Forestry, Technical University of British Columbia, Vancouver BC, Canada
[5] Instituto Superior de Agronomia, Centro de Estudos Florestais, Universidade de Lisboa, Lisboa, Portugal
[6] Chair of Forest Growth and Dendroecology, Albert-Ludwigs-University, Freiburg, Germany
[7] Faculdade de Ciências—cE3c, Centre for Ecology, Evolution and Environmental Changes, Universidade de Lisboa, Lisboa, Portugal

Corresponding author
Augusta Costa,
augusta.costa.sousa@gmail.com,
augusta.costa@iniav.pt

## ABSTRACT

As climate change increasingly affects forest ecosystems, detailed understanding of major effects is important to anticipate their consequences under future climate scenarios. The Mediterranean region is a prominent climate change hotspot, and evergreen cork oak (*Quercus suber* L.) woodlands are particularly climatically sensitive due to cork (bark) harvesting. Cork oak's drought avoidance strategy is well-known and includes structural and physiological adaptations that maximise soil water uptake and transport and limit water use, potentially leading to reduced stem and cork growth. Trees' responses to cope with water-limited conditions have been extensively described based on cork-rings width and, more recently, on cork-rings density, in dendroecological studies. However, so far, tree functional attributes and physiological strategies, namely photosynthetic metabolism adjustments affecting cork formation, have never been addressed and/or integrated on these previous cork-rings-based studies. In this study, we address the relation between carbon and oxygen stable isotopes of cork rings and precipitation and temperature, in two distinct locations of southwestern Portugal–the (wetter) Tagus basin peneplain and the (drier) Grândola mountains. We aimed at assessing whether the two climatic factors affect cork-ring isotopic composition under contrasting conditions of water availability, and, therefore, if carbon and oxygen signatures in cork can reflect tree functional (physiological and structural) responses to stressful conditions, which might be aggravated by climate change. Our results indicate differences between the study areas. At the drier site, the stronger statistically significant negative cork $\delta^{13}C$ correlations were found with mean temperature, whereas strong positive cork $\delta^{18}O$ correlations were fewer and found only with precipitation. Moreover, at the wetter site, cork rings are enriched in $^{18}O$

and depleted in $^{13}$C, indicating, respectively, shallow groundwater as the water source for physiological processes related with biosynthesis of non-photosynthetic secondary tissues, such as suberin, and a weak stomatal regulation under high water availability, consistent with non-existent water availability constrains. In contrast, at the drier site, trees use water from deeper ground layers, depleted in $^{18}$O, and strongly regulate stomatal conductance under water stress, thus reducing photosynthetic carbon uptake and probably relying on stored carbon reserves for cork ring formation. These results suggest that although stable isotopes signatures in cork rings are not proxies for net growth, they may be (fairly) robust indicators of trees' physiological and structural adjustments to climate and environmental changes in Mediterranean environments.

# INTRODUCTION

Evergreen cork oak (*Quercus suber* L.) forests cover about 2.2 million hectares in the western Mediterranean basin region (*EUFORGEN, 2019*) and cork is the sixth-most important non-wood forest product globally (*FAO, 2014*). Located in one of the most vulnerable regions of the globe (*IPCC , 2017*), these forests are currently experiencing a great impact on their growth and cork yield (*Mendes et al., 2019*; *Costa & Cherubini, 2021*) driven by climate change, lower precipitation, increased summer temperatures and more frequent and severe droughts (*Garzón, Dios & Ollero, 2008*; *Cherubini, Battipaglia & Innes, 2021*).

Cork oak produces narrow and ill-defined wood rings, contrasting with wide and sharp cork rings that are enhanced by cork harvesting (*Pereira, 2007*). So far, numerous dendroecological studies based on the width (*Caritat, Gutiérrez & Molinas, 2000*; *Costa et al., 2015*; *Oliveira, Lauw & Pereira, 2016*; *Ghalem et al., 2018*) and, more recently, on the chemical composition (*Leite et al., 2020*) and on the density of cork rings (*Costa et al., 2022*), have contributed to improve our knowledge on cork growth responses to inter-annual climate variability. However, despite this increased understanding of trees (and mainly cork) growth responses, these have scarcely been integrated with information on tree responses to water stress and/or under adverse environmental conditions. Only recently, the species' drought-avoidance strategy to cope with summer droughts (*Kurz-Besson et al., 2006*) was reported to affect cork growth responses, exacerbated by the pressure of the cork harvesting and under stressful environmental conditions (*Costa et al., 2016*; *Mendes et al., 2016*).

Stable isotopes in tree rings are widely applied in dendroecological studies to better understand tree physiological responses to environmental changes in temperate regions (*Lebourgeois, 2000*; *Ferrio et al., 2003*; *Saurer et al., 2008*; *Altieri et al., 2015*). Although stable isotopes do not indicate net growth, they reflect tree functional attributes and physiological processes, namely photosynthetic metabolism adjustments, to cope with

climate variations, under specific environmental conditions (*Loader et al., 2008*). Apart from differences and changes in the stable isotopic composition of different potential sources (*e.g.*, $\delta^{13}C$ of atmospheric $CO_2$, for carbon or $\delta^{18}O$ of water sources, for oxygen), the ratios of stable carbon and oxygen isotopes in tree rings are valuable to retrieve information on tree's intrinsic water-use efficiency ($\delta^{13}C$) and soil water sources ($\delta^{18}O$) (*O'Leary, 1988*; *Araguás-Araguás, Froehlich & Rozanski, 2000*; *Scheidegger et al., 2000*; *Loader, Robertson & McCarroll, 2003*; *McCarroll & Loader, 2004*; *Kurz-Besson et al., 2006*; *Ferrio et al., 2015*; *Hafner et al., 2015*).

Extending stable isotope analysis to cork rings appears a very attractive way to go beyond variability in net cork (tree) growth (*Oliveira, Lauw & Pereira, 2016*; *Ghalem et al., 2018*; *Leite et al., 2020*; *Costa & Cherubini, 2021*; *Costa et al., 2022*) and to address, in an integrative way, the physiological and photosynthetic metabolism adaptation of cork oak to climate changes. To our knowledge, such an approach has not yet been made and no studies on the stable isotopic composition of cork are available. We aimed to address this issue and postulated that, similarly to tree-ring-based (using wood or cellulose) studies, cork-ring-based (using cork or suberin) dendroecological assessments through stable isotope ratios would indicate differential adaptations of trees to local water-limited environmental conditions under Mediterranean climate. Suberin is the most important structural component of cork cell walls (*Graça & Pereira, 2000*) and its role is thus comparable to cellulose in wood cell walls. However, suberin is fundamentally different from cellulose (or lignin) as it is a lipid-based polymer (*Graça, 2015*) and, therefore, distinct biochemical effects during cellulose and suberin synthesis might occur affecting its carbon and oxygen isotope ratios, both still unknown in cork oak.

In our exploratory study on cork rings, we hypothesize that in contrasting soil water availability environments, the variation of $\delta^{13}C$ in the non-photosynthetic tissues such as cork reflects tree adjustments in stomatal conductance and a biochemical impairment of photosynthetic rates due to increased water scarcity during the hot and dry summer. In turn, the variation of cork-ring $\delta^{18}O$ ratios would indicate trees physiological adaptation to environmental conditions, enabling the use of distinct water sources in the pathways of secondary tissues metabolism leading to suberin (and cork) formation (*Barbour, Walcroft & Farquhar, 2002*; *Barbour, 2007*; *Sternberg & Ellsworth, 2011*; *Song, Clark & Helliker, 2014*). To test these hypotheses, we analyzed cork samples from two locations where trees have potentially divergent and site-specific strategies to cope with summer drought: a peneplain site, where trees are able to reach shallow groundwater during summer (*Mendes et al., 2016*) and grown continuously, with no summer rest period (*Costa, Pereira & Oliveira, 2002*), and a hilly site where summer water stress leads to a drought-imposed rest, and trees eventually stop growing in a drought-avoidance strategy (*Cherubini et al., 2003*; *Costa et al., 2022*).

# SURVEY METHODOLOGY

## Study areas

In this study, we collected data from cork rings of selected trees in two distinct cork oak woodlands located in southwestern Portugal: Benavente (CL) (38°49′N, 8°49′W, 20

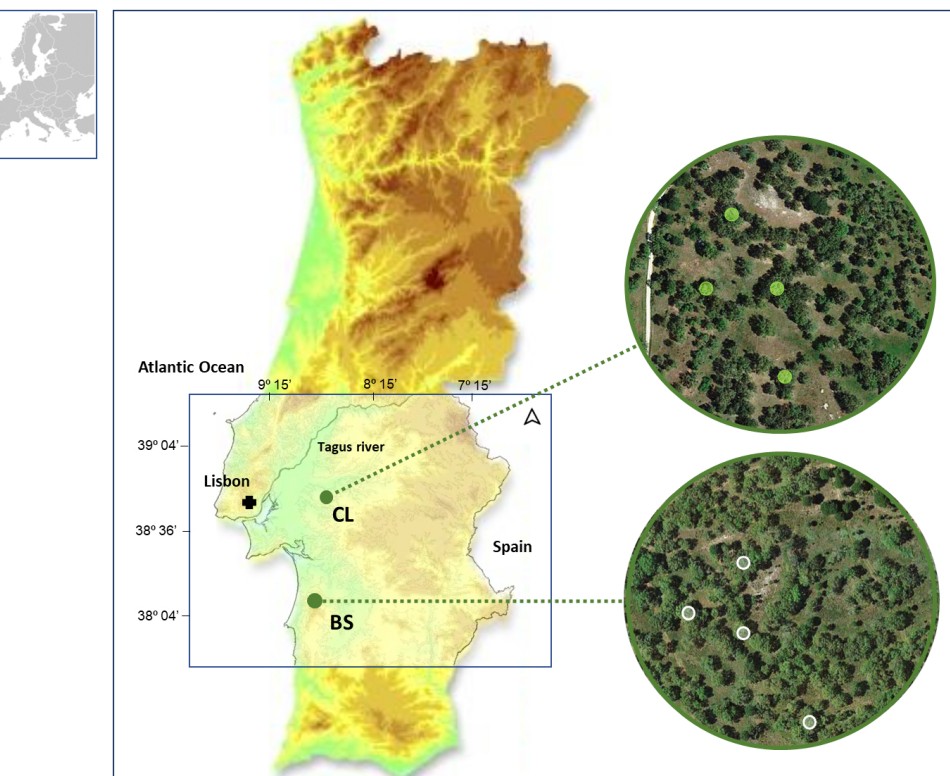

**Figure 1** **Location of the two study areas in SW Portugal, at Benavente (CL) and Grândola (BS).** Colored Contour areas indicate elevation classes, at gaps of 50 m, between 10 m a.s.l. (*e.g.*, in CL, in green) and over 250 m a.s.l. (*e.g.*, in BS, in light brown). In the image composition, aerial images are showing both cork oak woodlands with the sampled trees highlighted.

m a.s.l.) and ; Grândola (BS) (38°11′N, 8°37′W, 270 m a.s.l.) (Fig. 1). These cork oak woodlands have been described by the authors in their previous works (*Costa et al., 2022*; *Costa et al., 2016*).

Soil type, geologic formations and related biophysical site characteristics of these areas (CL and BS), fully described in previous work (*Costa, Pereira & Oliveira, 2002*; *Costa et al., 2016*; *Costa et al., 2020*; *Mendes & Ribeiro, 2010*; *Mendes et al., 2016*; *Mendes et al., 2019*), have distinctly affected soil water availability, specifically during the dry and hot summer season. At CL, trees have access to shallow groundwater, the main reliable water source for the maintenance of their physiologic activity during summer (*Costa, Pereira & Oliveira, 2002*). In contrast, trees at BS have a summer dormancy period (*Oliveira et al., 1994*) that is a drought stress-imposed rest period (*Cherubini et al., 2003*). At BS, tree growth must rely on groundwater that it is only used when the water table is reachable by tree roots, in early spring (*Costa, Madeira & Oliveira, 2008*; *Costa et al., 2016*).

Both study areas have a Mediterranean climate characterized by dry and hot summers and rainy winters fully described in *Costa et al. (2022)*.

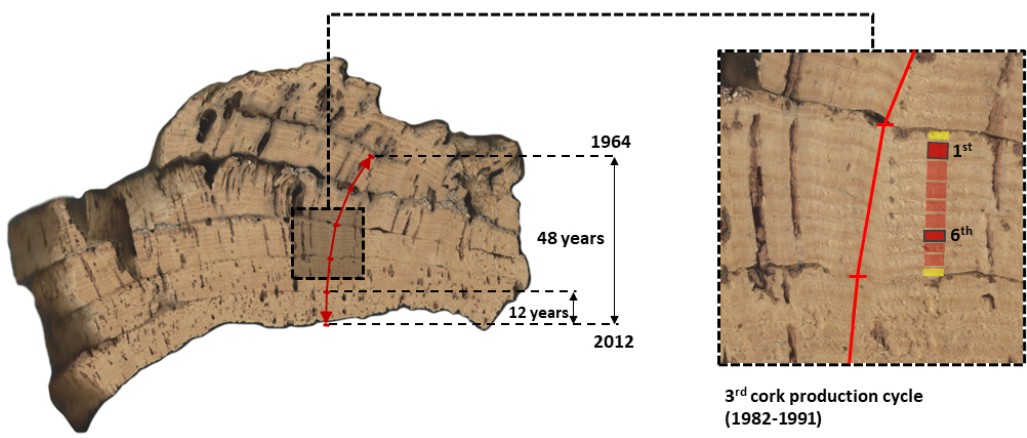

**Figure 2** Image of a cross-section of one multi-layered cork sample from CL (Benavente) showing the five consecutive cork production cycles that were used in this study to extract and measure long-term series of reproduction cork rings (1964–2012). This cork sample (c. 15–30 cm) was collected in the tree stem at the (maximum cork harvesting) height of 1.80 m above ground. The considered five cork production cycles (1st to 5th, between 1964 and 2012) are identified and clearly visible, each encompassing 9 years, except for the most recent one, with 12 years (2000–2012). In the image composition, the third cork production cycle (1982–1991) was amplified showing their cork-rings with the first and sixth cork-rings highlighted.

Precipitation and temperature raw data were obtained from the SNIRH (*Sistema Nacional de Informação de Recursos Hídricos*) database and were computed to obtain series of climate parameters for the period 1962-2012. Precipitation and temperature were calculated monthly, from February of the previous year to November of the current growth year. One (monthly), five- and twelve-months response functions were established for cork-rings $\delta^{13}$C, $\delta^{18}$O, width ($C_{RW}$) and maximum density ($C_{MD}$).

## Cork ring $\delta^{13}$C and $\delta^{18}$O data collection

Cork samples were collected from four healthy trees at each site (CL and BS) during the cork harvesting seasons (June) of 2012 and 2013. Such cork samples are complex, multi-layered samples, encompassing reproduction cork-rings that were induced by multiple and consecutive cork harvests (Fig. 2). Thus, these samples allow addressing long-term chronologies of cork rings in individual trees (*Costa et al., 2015*; *Costa et al., 2022*). In each sample, cork rings were visually dated using image analysis techniques (*Costa et al., 2015*; *Costa & Cherubini, 2021*) for the period 1962–2013.

Generally, cork production cycles at the mountain site (BS) are longer (10 years) than at the peneplain site (CL) (9 years) (*Costa et al., 2022*). At BS, cork samples were harvested in 2012, all comprised five consecutive cork production cycles of 10 years, initiated in 1962 (first cork production cycle), 1972, 1982, 1992 and 2002 (fifth cork production cycle), with the final cork harvest in 2012. At CL, cork samples per tree were harvested in 2013, to which corresponded five consecutive cork production cycles of 9 years starting in 1968 (first), 1977, 1986, 1995 and 2004 (fifth), and harvested in 2012 to which corresponded five

cork production cycles initiated in 1964 (first), 1973, 1982, 1991 and 2000 (exceptionally, the fifth cycle lasted 12 years) (Fig. 2).

Cork-ring width ($C_{RW}$) and maximum density ($C_{MD}$) chronologies were generated for each cork sample (*Costa et al., 2022*). Since $C_{RW}$ is heavily influenced by cork harvesting, each cork-ring width series was detrended by fitting a polynomial function (*Costa, Pereira & Oliveira, 2002*). Cork-ring width indices (residuals) ($IC_{RW}$) were calculated by dividing the observed $C_{RW}$ values by the predicted ones and $IC_{RW}$ chronologies were then built. On the other hand, $C_{MD}$ chronologies did not seem to contain any cork harvesting related trend (*Costa et al., 2022*) and therefore did not require any statistical detrending. Cork-ring maximum density indices ($IC_{MD}$) were calculated by dividing the observed $C_{MD}$ values by their mean value in each cork production cycle and, this way the $IC_{MD}$ chronologies were built.

Due to technical constrains to obtain enough cork mass for isotopic analysis, not all cork rings could be analyzed. In all samples, we selected the 1st and the 6th rings, respectively with the cork age ($y_{cork}$) equal to 1 and 6, of each of the considered five production cycles. The 1st ring is the largest one, representative of the trees response to cork harvesting and the 6th ring is a middle cycle ring, representative of an average cork growth, in the absence of stressful disturbances (Fig. 2). Data on carbon and oxygen isotopes were thus ascribed to both cork ages in all the consecutive cycles, regardless of the time series. By selecting fixed years (1st and 6th cork rings) of each cork-production cycle we expected to obtain information reflecting the intra- and inter-tree variability in tree physiological responses to drought conditions, under the influence of cork harvesting. A total of 78-records for ($\delta^{13}C$) and for ($\delta^{18}O$) were collected in the cork-rings of the study areas, 44 records for BS and 34 for CL.

Stable isotope ratio analyses were performed at the Stable Isotopes and Instrumental Analysis Facility (SIIAF) of the Centre for Ecology, Evolution and Environmental Changes (cE3c), University of Lisbon, Portugal. The stable-carbon isotope ratio ($^{13}C/^{12}C$) of cork was measured by continuous flow isotope mass spectrometry (CF-IRMS), on a Sercon Hydra 20-22 (Sercon, Hook, Hampshire, UK) stable isotope ratio mass spectrometer, coupled to a EuroEA (EuroVector, Italy) elemental analyzer for online sample preparation by Dumas-combustion. The stable-oxygen isotope ratio ($^{18}O/^{16}O$) of cork and suberin was also determined by continuous flow isotope mass spectrometry (CF-IRMS), coupled to a high temperature pyrolysis unit (HT-PyrOH, Eurovector, Italy) and a EuroEA3000 (EuroVector, Lombardy, Italy) elemental analyzer for online sample preparation by thermal degradation (thermolysis). The precision of the isotope ratio analysis, calculated using values from six to nine replicates of standard material interspersed among samples in every batch analysis, was $\leq 0.2‰$. Results are given in the $\delta$-notation, *i.e.,* in the delta notation relative to the international standards:

$$\delta sample = (Rsample/Rstandard - 1) \times 1{,}000,$$

where Rsample is the molar fraction of $^{13}C/^{12}C$, or $^{18}O/^{16}O$ ratio of the sample and Rstandard, of the standards of Vienna Pee Dee Belemnite (VPDB) for carbon and Vienna Standard Mean Ocean Water (VSMOW) for oxygen.

## Statistical analyses

The climate-isotope relationships were tested based on monthly Pearson correlations considering one month and five- and twelve-months windows, from February of the prior growth year till November of the current growth year. We included prior-year climate variables in the analysis because climate in the preceding growing season often influences stable isotope values in the following season due to carbon carry-over effects (*Gessler et al., 2014*).

Linear mixed-effects modelling was used to examine the influence of the covariate cork age ($y_{cork}$) at the two sites (non-water-limited area, CL and water-limited area, BS) on the isotope-derived physiological responses of cork oak and; to examine the variability of the cork $\delta^{13}$C and $\delta^{18}$O relationships with climate parameters, precipitation and temperature. Climate variables were thus considered as the fixed factors for the response functions $\delta^{13}$C and $\delta^{18}$O composition of cork rings. Also, linear mixed-effects models would explicitly integrate among-tree variability, considering trees as a random effect, related with tree responses to limited water availability. Model fitting was done by adding random effects of site $Xy_{cork}$ (four classes) and tree (8 classes), through random intercept alone. We evaluated model performance using the Akaike information criterion (AIC). The model with the lowest AIC value was considered the best to predict response variables ($\delta^{13}$C and $\delta^{18}$O). We further evaluated the variation explained by the random and fixed effects together by calculating marginal $R^2$ to define the error or statistical uncertainty related with the inter-tree variability. All the analyses were conducted in R statistical environment (*R Development Core Team, 2019*) using the *nlme* package in R.

## RESULTS

### $\delta^{13}$C composition in cork rings and its correlations with climate

The mean isotopic compositions ($\delta^{13}$C) of corkrings ranged between $-27.3$‰ at CL and $-26.8$‰ at BS (Fig. 3). Average $\delta^{13}$C values at BS indicated that here cork was slightly more enriched in $^{13}$C (over 0.5‰) than at CL. However, there was no significant difference in cork-rings $\delta^{13}$C between the trees from BS and CL (nested ANOVA results showed no significant differences between the two groups (sites), $F(1, 6) = 0.676$, *p*-value $= 0.442$). Between trees within each study area, the variations were statistically significant (nested ANOVA results showed $F(6, 70) = 6.951$, *p*-value $= 0.000$). At CL, the cork of tree CL44 differed significantly from the other trees and; at BS, the cork of trees BS1 and BS8 differed significantly from the other two trees. CL44, BS1 and B8 were distinctly enriched in $^{13}$C (less negative $\delta^{13}$C values) when compared to the other trees (Fig. 3).

Furthermore, in all cork samples, except for the cork of tree CL44 (at CL), the mean isotopic compositions ($\delta^{13}$C) of 1st cork-rings ($y_{cork} = 1$) were less negative than 6th cork-rings ($y_{cork} = 6$) which indicates that, on average, cork formed immediately after cork harvesting is more enriched in $^{13}$C when compared to the cork formed later, hereafter the 6th year in the cork harvesting cycle.

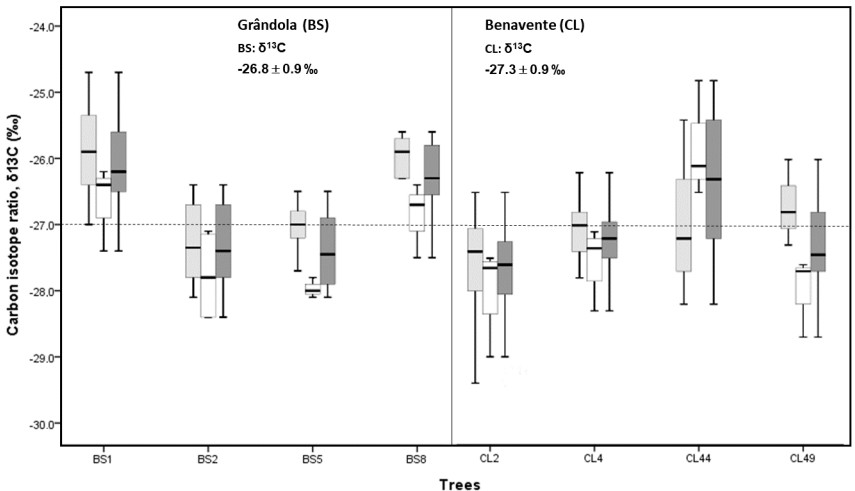

**Figure 3** **Variability of $\delta^{13}$C in cork-rings ($n = 78$) of the four trees from the two study areas, Grân-dola ($n = 44$) (BS) (in the left) and Benavente ($n = 34$) (CL) (in the right).** Solid light grey boxes for $y_{cork}$ = 1 (1st cork growth years) ($n = 24$ for BS and $n = 22$ for CL); solid white boxes for $y_{cork} = 2$ (6th cork growth years) ($n = 20$ for BS and $n = 12$ for CL) and solid dark grey boxes for all cork growth years. Multi-boxplots (median; box, interquartile range: IQR = Q75percent − Q25percent; whiskers, minimum and maximum). For each study area, mean ± standard deviation of $\delta^{13}$C in cork-rings. Dotted line is for aver-age value of cork $\delta^{13}$C of −27.0 ‰.

The Pearson's correlations (r) between $\delta^{13}$C of cork rings and mean temperature showed similarities among the trees of CL and BS, and were significantly negative (Figs. 4A–4B). This is a striking result which suggests that the $\delta^{13}$C (‰) of cork rings will be lower in warmer years, *i.e.,* cork will be more $^{13}$C depleted.

Noticeably, besides a few significant monthly correlations (Fig. 4A), the relationships became more robust after grouping temperature variables in periods of 5- or 12-months, mainly at BS (Fig. 4B). The variation of cork $\delta^{13}$C at BS was much more sensitive to temperature than at CL, with a marked sensitivity to prior conditions since spring of the previous year throughout the growth year (Fig. 4B). The highest negative correlation coefficients were found for the mean temperatures previous to the growth year, in February–September ($r = -0.423$, $p$-value = 0.004) at BS and in April–May ($r = -0.462$, $p$-value = 0.006) at CL. Moreover, at BS, $\delta^{13}$C also correlated with temperature of the current growing season, in the spring period (March = −0.371, $p$-value = 0.013) (Figs. 4A–4B), in the onset of the cork growth flush.

Weaker correlations were found between $\delta^{13}$C of cork rings and precipitation in both study areas (Figs. 4C–4D). The precipitation of current-year March (at CL) and June (at BS) positively influenced cork $\delta^{13}$C (Fig. 4C). In contrast with temperature, no significant correlations were found with grouped precipitation values (periods of 5- to 12-months) (Fig. 4D). A relatively weak correlation was observed only at BS, in the summer of the current growth year in June–August ($r = 0.350$; $p$-value = 0.021). Also, in contrast with temperature, results revealed one irregularity of the relationship between cork $\delta^{13}$C and

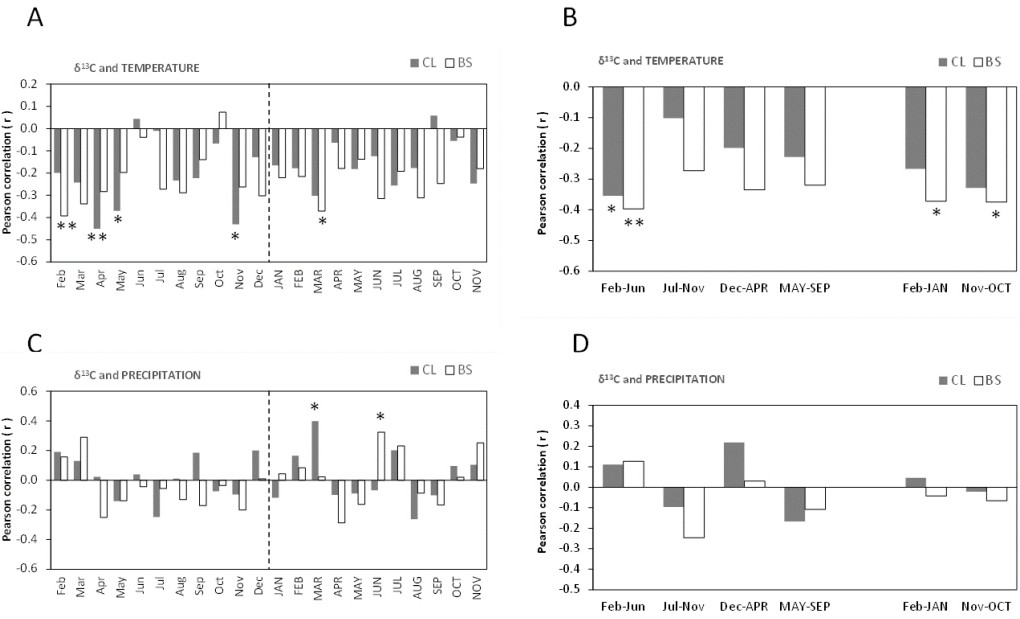

**Figure 4** Pearson correlation coefficient ($r$) between $\delta^{13}$C of all cork rings from Grândola ($n = 44$) (BS) and Benavente ($n = 34$) (CL) and: (A) Monthly mean temperature from February previous to the current growth year (Feb) to November of current growth year (NOV). Significant correlations were found for BS in: Feb ($r = -0.393^{**}$, $p$-value = 0.008) and MAR ($r = -0.371^*$, $p$-value = 0.013) and, for CL in: Apr ($r = -0.451^{**}$, $p$-value = 0.008); May ($r = -0.370^*$, $p$-value = 0.031) and in Nov ($r = -0.431^*$, $p$-value = 0.011). (B) Selected five- and twelve-months mean temperature. Significant correlations were found for BS in: Feb–Jun ($r = -0.397^{**}$, $p$-value = 0.008); in Feb–JAN ($r = -0.372^*$, $p$-value = 0.013) and in Nov–OCT ($r = -0.375^*$, $p$-value = 0.013) and, for CL in: Feb–Jun ($r = -0.355^*$, $p$-value = 0.039). (C) Monthly precipitation. Significant correlations were found for BS in: JUN ($r = 0.324^*$, $p$-value = 0.032) and for CL in MAR ($r = 0.398^*$, $p$-value = 0.020); (D) Selected five- and twelve-months precipitation. Asterisks represent significant correlations (*, $p$-value < 0.05 and **, $p$-value < 0.01). Lower case, months of year prior to the growth year; capitals, months of current growth year.

precipitation: although not statistically significant, correlations were either positive or negative correlations, regardless of the considered periods (Fig. 4D).

## $\delta^{18}$O composition in cork rings and its correlations with climate

Cork-ring $\delta^{18}$O averaged $21.3 \pm 0.9\permil$ (CL) and $20.9 \pm 0.9\permil$ (BS). Average $\delta^{18}$O values indicate that at CL, cork was enriched in $^{18}$O, over $0.4\permil$ more than at BS (Fig. 5). However, there was no significant difference in cork-rings $\delta^{18}$O between the trees from BS and CL (nested ANOVA results showed no significant differences between the two groups (sites), $F(1, 6) = 3.188$, $p$-value = 0.121). Between trees within each study area, the variations were not statistically significant (nested ANOVA results showed $F(6, 70) = 1.989$, $p$-value = 0.079). Furthermore, mean values of suberin's $\delta^{18}$O in cork rings ranged between $16.8\permil$ at CL and $16.3\permil$ at BS, in accordance with the values found for cork.

At CL, the cork of tree CL4 differed significantly from all other trees, and it was depleted in $^{18}$O when compared to the other trees (Fig. 5).

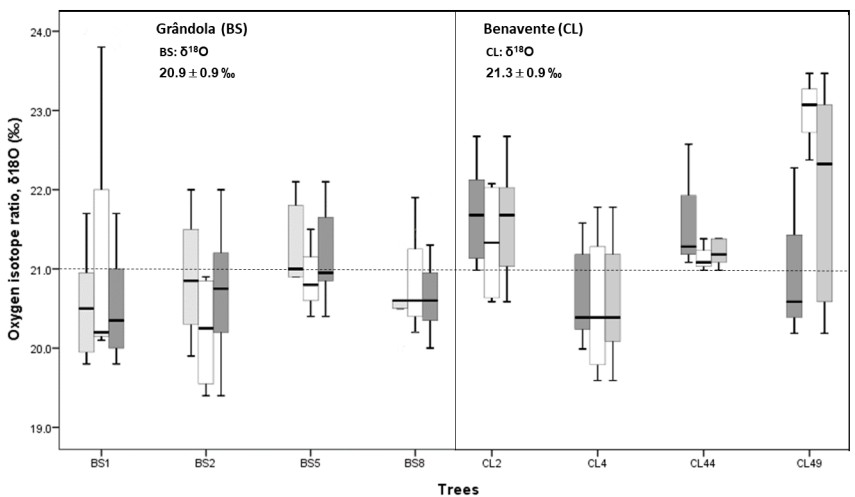

**Figure 5** Variability of cork stable isotope ratios, $\delta^{18}$O in cork-rings ($n = 78$) of the four trees from the two study areas, Grândola ($n = 44$) (BS) (in the left) and Benavente ($n = 34$) (CL) (in the right). Solid light grey boxes for $y_{cork} = 1$ (1st cork growth years) ($n = 24$ for BS and $n = 22$ for CL); Solid withe boxes for $y_{cork} = 2$ (6th cork growth years) ($n = 20$ for BS and $n = 12$ for CL) and solid dark grey boxes for all cork growth years. Multi-boxplots (median; box, interquartile range: IQR = Q75percent − Q25percent; whiskers, minimum and maximum). For each study area, mean ± standard deviation of $\delta^{18}$O in cork-rings. Dotted line is for average value of cork $\delta^{18}$O of 21.0 ‰.

In all cork samples, except for the cork of tree CL49 (at CL), the mean isotopic compositions ($\delta^{18}$O) of 1st cork-rings ($y_{cork} = 1$) were slightly higher or equal than 6th cork-rings ($y_{cork} = 6$) which indicates that cork formed immediately after cork harvesting is somewhat more enriched in $^{18}$O when compared to the cork formed later, in the 6th year after the cork harvesting (Fig. 6).

The oxygen isotope signal in cork rings is more sensitive to precipitation than to temperature (Fig. 6). The correlations between $\delta^{18}$O and mean temperature were statistically significant only in trees at CL (Figs. 6A–6B). At this site, we found negative correlations with the temperature only in spring prior to the growth year. Besides the weak isolated monthly correlation with May ($r = -0.345$, $p$-value = 0.046) (Fig. 6A), this relationship became slightly more robust when after grouping temperature shows statistically significant correlations, in the summer prior to the growth year, between May–June ($r = -0.373$, $p$-value = 0.030) and May–August ($r = -0.359$, $p$-value = 0.037).

The correlations found between $\delta^{18}$O and precipitation were weak and negligible at BS but not at CL (Figs. 6C–6D). Here, $\delta^{18}$O relationship was positive and statistically significant with precipitation in the current spring, in March–May ($r = 0.497$, $p$-value = 0.003) and in spring-early summer, in March–June ($r = 0.428$, $p$-value = 0.012).

## Correlations between width and density of cork-rings and climate

At the two study areas, none of the correlations between cork-ring isotopic compositions ($\delta^{13}$C and $\delta^{18}$O), cork-ring width (IC$_{RW}$) and cork-ring maximum density (IC$_{MD}$) were statistically significant.

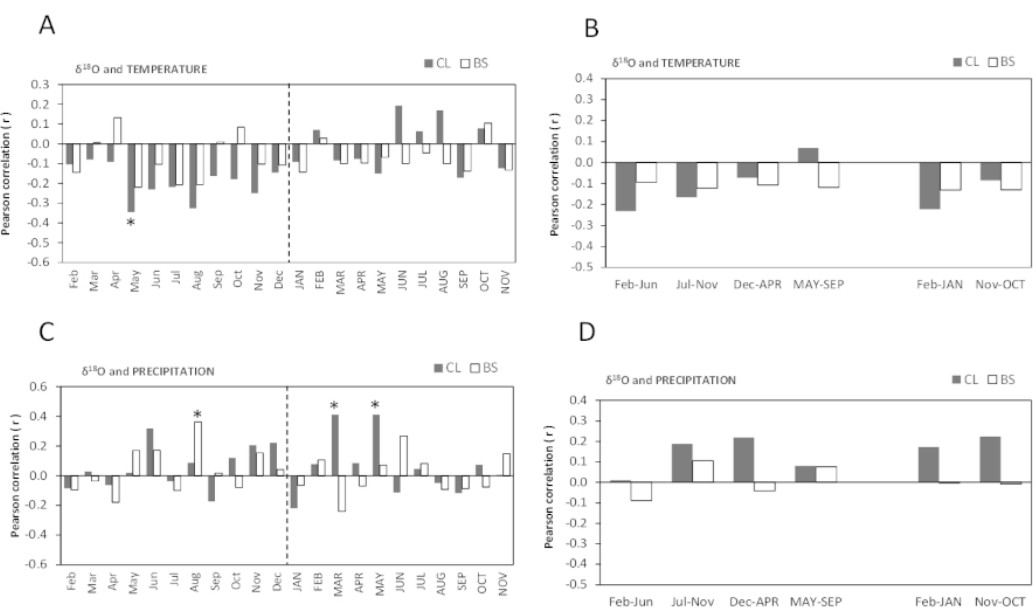

**Figure 6** Pearson correlation coefficient ($r$) between $\delta^{18}$O of all cork rings from Grândola ($n = 44$) (BS) and Benavente ($n = 34$) (CL) and: (A) Monthly mean temperature from February previous to the current growth year (Feb) to November of current growth year (NOV). Significant correlations were found for CL in: May ($r = -0.345^*$, $p$-value = 0.046); (B) Selected five- and twelve-months mean temperature; (C) Monthly precipitation. Significant correlations were found for BS in: Aug ($r = 0.362^*$, $p = 0.016$) and for CL in MAR ($r = 0.411^*$, $p$-value = 0.016) and in MAY ($r = 0.411^*$, $p$-value = 0.016); (D) Selected five- and twelve-months precipitation. Asterisks represent significant correlations ($^*$, $p$-value < 0.05). Lower case, months of year prior to the growth year; capitals, months of current growth year.

The two study areas clearly differ in the relation between $IC_{RW}$ and climate parameters. At BS, the results are inconsistent regarding precipitation: (i) within the year prior to the growth year, two distinct sub-periods: one in the spring (March–June), positively correlated with precipitation, and another in summer-autumn (June–October), negatively correlated and; (ii) in the current growth year, contrasting weaker positive correlations in summer-autumn (July–September) (Table 1). Here, despite the non-significant relation between $IC_{RW}$ and mean temperature of the current growth year (in line with the non-significant relation with $IC_{MD}$, Table 1), the mean temperature within March and November prior to the growth year influenced cork growth much more than the cork density (Table 1).

At CL, an irregular pattern of correlations between ICRW and climate parameters was observed: negative correlations between precipitation and $IC_{RW}$, mainly in July–November of the current growth year, striking, strong and positive correlations between mean temperature in spring (April–May) of the year prior to the growth year, and negative correlation with mean temperature in the winter prior to the growth year (December–February) (Table 1).

Relationships between climate parameters (precipitation and temperature) and $IC_{MD}$ were only relevant at BS; at CL the correlations were statistically not significant (Table 1). At former site, precipitation conditions prior to the growth year negatively affected $IC_{MD}$,

**Table 1  Selected significant correlation coefficients ($r$) between $IC_{RW}$ and $IC_{MD}$ and precipitation and mean temperature, of all cork rings from Benavente ($n = 40$) (CL) and Grândola ($n = 52$) (BS).**

| Parameter | Study area | Precipitation | | Temperature | |
|---|---|---|---|---|---|
| | | Months | $r$ | Months | $r$ |
| $IC_{RW}$ | CL | February$_{(-1)}$ | −0.331* | April$_{(-1)}$ | 0.349* |
| | | | | April$_{(-1)}$–May$_{(-1)}$ | 0.376* |
| | | | | December$_{(-1)}$–February | −0.409** |
| | | July–November | −0.329* | | |
| | BS | March$_{(-1)}$–June$_{(-1)}$ | 0.297* | March$_{(-1)}$–September$_{(-1)}$ | 0.277* |
| | | May$_{(-1)}$ | 0.307* | April$_{(-1)}$–September$_{(-1)}$ | 0.284* |
| | | June$_{(-1)}$–October$_{(-1)}$ | −0.348* | May$_{(-1)}$–November$_{(-1)}$ | 0.281* |
| | | September$_{(-1)}$–October$_{(-1)}$ | −0.342* | August$_{(-1)}$–September$_{(-1)}$ | 0.279* |
| | | October$_{(-1)}$ | −0.323* | | |
| | | July–September | 0.298* | | |
| | | September | 0.286* | | |
| | | September–October | 0.283* | | |
| $IC_{MD}$ | CL | | n.s. | | n.s. |
| | BS | May$_{(-1)}$–October$_{(-1)}$ | −0.275* | May$_{(-1)}$ | 0.278* |
| | | May | 0.273* | | |
| | | August | 0.329* | | |

Notes.

Significant levels according to Student's test: * $p$ value < 0.05; ** $p$ value < 0.01.

(−1) indicate the months in the year prior to the growth year.

from May to October. Conversely, the $IC_{MD}$ correlations with precipitation of current growth year May and August were statistically significant and positive. Moreover, between temperature and $IC_{MD}$, only a relative weak significant correlation was found, with mean temperature of May prior to the growth year ($r = 0.278$, $p$-value $= 0.046$) (Table 1).

The correlations found between mean temperature and $IC_{RW}$ and $IC_{MD}$ at BS, and between precipitation and $IC_{RW}$ and $IC_{MD}$ at CL, were considerably weaker than the correlations found with $\delta^{13}C$ (BS) and with $\delta^{18}O$ (CL).

### Influence of cork harvesting and site on the relation between climate and cork- rings isotope composition ($\delta^{13}C$ and $\delta^{18}O$)

Selected mixed-effects linear models (M0–M5) were plotted against the site $Xy_{cork}$ and trees random-effects (Table 2). The fixed-effect variables were the climate variables which displayed the highest bootstrap correlation coefficients. For $\delta^{13}C$ were considered the mean temperatures of: February$_{(-1)}$–June$_{(-1)}$ (M1) ($r = -0.355$, $p$-value $= 0.039$ for CL and $r = -0.397$, $p$-value $= 0.008$ for BS); April$_{(-1)}$–May$_{(-1)}$ (M2) ($r = -0.462$, $p$-value $= 0.006$ for CL and $r = -0.303$, $p$-value $= 0.013$ for BS) and; March (M3) ($r = -0.302$, $p$-value $= 0.082$ for CL and $r = -0.371$, $p$-value $= 0.013$ for BS). For $\delta^{18}O$ were considered the precipitation of: August$_{(-1)}$ (M4) ($r = 0.087$, $p$-value $= 0.624$ for CL and $r = 0.362$, $p$-value $= 0.016$ for BS) and; March–June (M5) ($r = 0.428$, $p$-value $= 0.012$ for CL and $r = -0.220$, $p$-value $= 0.157$ for BS).

Costa et al. (2022), *PeerJ*, DOI 10.7717/peerj.14270

**Table 2** Coefficients, AIC values and R$^2$ for linear mixed-effects models of isotope ratios composition ($\delta^{13}$C and $\delta^{18}$O) of cork rings, with selected climatic variables as fixed effect and with site $Xy_{cork}$ and trees, as random factors.

Isotope composition ($\delta^{13}$C)

| Climate variables | MODEL | Model equation | $\alpha_0$ | $\alpha_1$ | $\sigma_{\text{sitexycorkj}}(\mu_j)$ | $\sigma_{\text{Treej}}(\mu_j)$ | $\sigma_{\varepsilon ij}$ |
|---|---|---|---|---|---|---|---|
| Temperature | | | | | | | |
| February$_{-1}$–June$_{(-1)}$ (**C**$_{\text{LIM1}}$) | M$_1$ | $\delta^{13}C_{ij} = \alpha_0 + \alpha_1 C_{\text{LIM1ij}} + \mu_{\text{sitexycorkj}} + \varepsilon_{ij}$ | −22.420 | −0.308$^{**}$ | 0.044 | | 0.713 |
| AIC | 206.06 | | | | | | |
| $R^2_{(\text{fixed effects})}$ | 0.12 | | | | | | |
| $R^2_{(\text{total})}$ | 0.17 | | | | | | |
| February$_{-1}$–June$_{(-1)}$ (**C**$_{\text{LIM1}}$) | M$_{10}$ | $\delta^{13}C_{ij} = \alpha_0 + \alpha_1 C_{\text{LIM1ij}} + \mu_{\text{Treej}} + \varepsilon_{ij}$ | −22.208 | −0.319$^{**}$ | | 0.274 | 0.467 |
| AIC | 184.94 | | | | | | |
| $R^2_{(\text{fixed effects})}$ | 0.13 | | | | | | |
| $R^2_{(\text{total})}$ | 0.45 | | | | | | |
| April$_{-1}$–May$_{(-1)}$ (**C**$_{\text{LIM2}}$) | M$_2$ | $\delta^{13}C_{ij} = \alpha_0 + \alpha_1 C_{\text{LIM2ij}} + \mu_{\text{sitexycorkj}} + \varepsilon_{ij}$ | −22.850 | −0.267$^{**}$ | 0.065 | | 0.696 |
| AIC | 205.08 | | | | | | |
| $R^2_{(\text{fixed effects})}$ | 0.12 | | | | | | |
| $R^2_{(\text{total})}$ | 0.20 | | | | | | |
| April$_{-1}$–May$_{(-1)}$ (**C**$_{\text{LIM2}}$) | M$_{20}$ | $\delta^{13}C_{ij} = \alpha_0 + \alpha_1 C_{\text{LIM2ij}} + \mu_{\text{Treej}} + \varepsilon_{ij}$ | −22.320 | −0.296$^{**}$ | | 0.308 | 0.438 |
| AIC | 181.12 | | | | | | |
| $R^2_{(\text{fixed effects})}$ | 0.15 | | | | | | |
| $R^2_{(\text{total})}$ | 0.50 | | | | | | |
| March (**C**$_{\text{LIM3}}$) | M$_3$ | $\delta^{13}C_{ij} = \alpha_0 + \alpha_1 C_{\text{LIM3ij}} + \mu_{\text{sitexycorkj}} + \varepsilon_{ij}$ | −24.163 | −0.232$^{**}$ | 0.031 | | 0.741 |
| AIC | 208.32 | | | | | | |
| $R^2_{(\text{fixed effects})}$ | 0.11 | | | | | | |
| $R^2_{(\text{total})}$ | 0.14 | | | | | | |
| March (**C**$_{\text{LIM3}}$) | M$_{30}$ | $\delta^{13}C_{ij} = \alpha_0 + \alpha_1 C_{\text{LIM3ij}} + \mu_{\text{Treej}} + \varepsilon_{ij}$ | −23.880 | −0.251$^{**}$ | | 0.284 | 0.472 |
| AIC | 185.95 | | | | | | |
| $R^2_{(\text{fixed effects})}$ | 0.13 | | | | | | |
| $R^2_{(\text{total})}$ | 0.45 | | | | | | |

**Table 2** (*continued*)

| Parameter | Study area | Precipitation | | | Temperature | | | | |
|---|---|---|---|---|---|---|---|---|---|
| | | Months | | $r$ | Months | $r$ | | | |
| Isotope composition ($\delta^{18}$ **O**) | | | | | | | | | |
| Climate variables | MODEL | Model equation | | $\alpha_0$ | $\alpha_1$ | $\sigma_{\text{sitexycorkj}}(\mu_j)$ | $\sigma_{\text{Treej}}(\mu_j)$ | | $\sigma_{\varepsilon ij}$ |
| Precipitation | | | | | | | | | |
| August$_{(-1)}$ ($\mathbf{C}_{\text{LIM4}}$) | $M_4$ | $\delta^{18}O_{ij} = \alpha_0 + \alpha_1 C_{\text{LIM4ij}} + \mu_{\text{sitexycorkj}} + \varepsilon_{ij}$ | | 20.914 | 0.029[*] | 0.002 | | | 0.876 |
| AIC | 218.99 | | | | | | | | |
| $R^2_{\text{(fixed effects)}}$ | 0.06 | | | | | | | | |
| $R^2_{\text{(total)}}$ | 0.06 | | | | | | | | |
| August$_{(-1)}$ ($\mathbf{C}_{\text{LIM4}}$) | $M_{40}$ | $\delta^{18}O_{ij} = \alpha_0 + \alpha_1 C_{\text{LIM4ij}} + \mu_{\text{Treej}} + \varepsilon_{ij}$ | | 20.959 | 0.027[*] | | 0.073 | | 0.807 |
| AIC | 217.63 | | | | | | | | |
| $R^2_{\text{(fixed effects)}}$ | 0.05 | | | | | | | | |
| $R^2_{\text{(total)}}$ | 0.13 | | | | | | | | |
| March–June ($\mathbf{C}_{\text{LIM5}}$) | $M_5$ | $\delta^{18}O_{ij} = \alpha_0 + \alpha_1 C_{\text{LIM5ij}} + \mu_{\text{sitexycorkj}} + \varepsilon_{ij}$ | | 20.960 | 0.001[*] | 0.005 | | | 0.917 |
| AIC | 220.29 | | | | | | | | |
| $R^2_{\text{(fixed effects)}}$ | 0.01 | | | | | | | | |
| $R^2_{\text{(total)}}$ | 0.01 | | | | | | | | |
| March–June ($\mathbf{C}_{\text{LIM5}}$) | $M_{50}$ | $\delta^{18}O_{ij} = \alpha_0 + \alpha_1 C_{\text{LIM5ij}} + \mu_{\text{Treej}} + \varepsilon_{ij}$ | | 21.000 | 0.008[*] | | 0.027 | | 0.835 |
| AIC | 218.32 | | | | | | | | |
| $R^2_{\text{(fixed effects)}}$ | 0.01 | | | | | | | | |
| $R^2_{\text{(total)}}$ | 0.10 | | | | | | | | |

**Notes.**

Significance of predictors is indicated by: * (*p*-value < 0.05) and ** (*p*-value < 0.01).

In all the models, $\delta^{13}$C in cork rings decreased with mean temperature and confirmed our previous results that $\delta^{13}$C of cork rings was lower in warmer years, *i.e.,* $^{13}$C depleted (Figs. 4A–4B). In the best model for $\delta^{13}$C (M2—with the lowest AIC of 205.08 and explaining about 20% of the total variability), $\delta^{13}$C decreased on average about $-0.267$‰ per °C of monthly spring temperature (April$_{(-1)}$–May$_{(-1)}$) (Table 2). Despite the general negative correlation, in this model the predicted values of $\delta^{13}$C in the cork-rings formed immediately after cork harvesting ($y_{cork} = 1$) at a drier site (BS) were clearly higher (less negative $\delta^{13}$C), with an $\alpha_0 = -22.482$, than in the later cork-rings ($y_{cork} = 6$) or at the CL wetter site (with either $y_{cork}$ of 1 or 6), with an $\alpha_0 = -23.004$, for the same negative slope ($\alpha_1 = -0.267$, $p < 0.001$).

Model M20 for $\delta^{13}$C, with trees as random effect, had the lowest AIC, 181.12 and explained about 50% of the total variation (Table 2). This suggests a most strong influence of between-tree variability on the variations of cork-rings $\delta^{13}$C with mean temperature, mainly when compared to model M2 that explained a non-negligible 20% of total variation and that the site $Xy_{cork}$ random effect explained up to 15% of total variability.

In the models with $\delta^{18}$O as the response function, $\delta^{18}$O increased with the monthly precipitation, in accordance with previous results on significant correlations (Figs. 6C–6D). The models M4 (with an AIC of 218.99 and explaining 6% of the total variation) and M5 (with an AIC of 220.29 and explaining 1% of the total variation), poorly explained the total variation. Furthermore, no variation was explained by the site $Xy_{cork}$ random effect (Table 2). However, in the random intercept model (M5), the predicted values of $\delta^{18}$O in cork-rings formed at a drier site (BS) were clearly different and slightly depleted in $^{18}$O (lower $\delta^{18}$O, with an $\alpha_0 = 20.772$), than in the wetter site (CL), with higher $\delta^{18}$O ($\alpha_0 = 21.039$). These results would suggest the relative higher influence of site on the depth of the preferred source water for cork rings formation, independently on the effect of the cork harvesting.

## DISCUSSION

### Influence of water availability conditions on the $\delta^{13}$C of cork rings

Cork oak has the adaptive capacity to cope with variations in soil water availability during the growing season and can adjust its annual growth rhythm to prevailing environmental constrains, under the highly seasonal Mediterranean climate conditions (*Cherubini et al., 2003*; *De Micco et al., 2016*). Trees at Grândola (BS) typically present a bimodal growth pattern, as growth is subjected to a double constraint, by low temperatures in winter, and by drought in summer as in temperate climates (*Cherubini et al., 2003*). Generally, the onset of cork (and tree) growth occurs in spring, after a relative cold and wet winter, when soil water reserves have been replenished. In the summer, with limited water availability, trees experience a drought stress period and stop their stem (and cork) radial growth in July–August (summer rest period), resuming growth in autumn. However, trees at Benavente (CL) are able to reach the groundwater (*Mendes et al., 2016*), and to maintain their growth uninterruptedly from March to November; in fact, it is during the summer (June–July) that cork oaks' radial growth (that it is mostly cork growth) reaches a maximum

peak at this site (*Costa, Pereira & Oliveira, 2003*). So, different physiological growth patterns would be expected between the two sites.

Under low water availability (at BS), the higher cork rings $\delta^{13}$C, *i.e.,* cork-rings enriched in $^{13}$C (on average, $-26.8‰$ at BS against $-27.3‰$ at CL) (Fig. 3) would be mostly explained by a strong regulation (reduction) of stomatal conductance, possibly affecting photosynthesis, confirming our first hypothesis on that the cork rings $\delta^{13}$C variation may reflect the physiological and structural adjustments (*e.g.,* stomatal conductance) to cope with drought. The former values are slightly lower but in line with the range of values between $-26.3‰$ and $-26.1‰$ found in tree rings of the evergreen oak *Q. ilex* (*Zalloni et al., 2018*), where wood formation is determined by a double climate stress (summer drought and winter cold) and usually results from two phases of cambial activity separated by the summer rest period. Since cork mass is 40–50% suberin, a lipid-based polymer (*Graça & Pereira, 2000*; *Graça, 2015*), our results on $^{13}$C depleted cork-rings in comparison with tree-rings are in accordance with previous studies reporting that the metabolic pathway involved in lipid synthesis is much more $^{13}$C depleted than the one involved in (wood) cellulose synthesis. This occurs due to the $^{13}$C depletion of primary assimilates in the leaves to produce lipids, in contrast with $^{13}$C enrichment to produce cellulose (*De Niro & Epstein, 1977*; *Melzer & Schmidt, 1987*; *Gessler & Ferrio, 2022*). Moreover, our results on the correlations of cork rings $\delta^{13}$C with climate variables confirmed the critical role of temperature. This agrees with the fact that trees are forced to concentrate their photosynthetic activity in the warmer months of spring and autumn at both sites but mainly at the drier one (BS) (Fig. 4 and Table 2).

Noticeably, there is an apparent paradox in our results as trees at both sites: CL and BS, showed a general decreasing trend of $\delta^{13}$C in cork rings (*i.e.,* $^{13}$C depleted cork rings) with increasing temperatures, and significant negative correlations between $\delta^{13}$C and mean temperature of the previous-year spring (Figs. 4A–4B). However, this depletion of $^{13}$C in cork rings can be explained by the possibility of the occurrence of post-photosynthetic processes associated with the remobilization and use of previous year(s) synthetized and stored carbon for cork formation, due to the shortage, or even the $^{13}$C discrimination of recent photo-assimilates in current (early) spring which are effectively $^{13}$C enriched (with higher $\delta^{13}$C) due to reduction of stomatal conductance. In cork oaks, a transfer of carbon reserves from the non-conducting to the conducting phloem tissue, close to the zone where phellogen is active, was reported by *Knapic et al. (2007)*. These authors speculated that a displacement of these reserves in the form of extractives from heartwood to the outer sapwood for cork formation occurs at expenses of wood growth. It is possible that isotopic fractioning occurs during transport of metabolites As according to *Cerasoli et al. (2004)*, the mobilization and utilization of stored carbon in cork oaks implies the hydrolysis of starch and the synthesis of sucrose, both enriched in $^{13}$C (*Eglin et al., 2010*) with a reduction of its concentration in stem wood. One can suggest that the preferential carbon reserves used for cork growth would be depleted in $^{13}$C, perhaps formed in less water stressed periods, and/or that the fractioning at the biochemical pathways of suberin synthesis would lead to compounds with very low $\delta^{13}$C (*Damesin & Lelarge, 2003*).

Trees at Benavente (CL) are with unlimited water availability and thus less water stressed. Cork growth should rely mainly on current photosynthates, relatively more depleted in $^{13}$C, in the onset of cork formation, in spring. Moreover, in autumn and warmer winters, trees use and probably store current photosynthates for the next season, which are less enriched in $^{13}$C (lower $\delta^{13}$C) given the lower tree water stress. In BS, relatively higher mean values of $^{13}$C in cork rings reflect limited water availability and trees water stress. Spring cork growth flush should rely not only on $^{13}$C enriched current photosynthates but on enriched $^{13}$C carbon reserves, which were stored during the previous dormant (winter) season (*Ghalem et al., 2018*), similarly to spring xylem (early wood) formed in tree-rings of other Mediterranean evergreen oaks, *e.g.*, *Q. ilex* (*Ferrio et al., 2003*; *Gea-izquierdo, Cherubini & Cañellas, 2011*; *Shestakova et al., 2014*; *Zalloni et al., 2018*), and even in deciduous *Q. petraea* (*Barbaroux & Bréda, 2002*).

Furthermore, suberin is the main structural compound of the cork cell walls and it has a very special biochemical composition, a polyester lipid build up from long chain (C16–C24) fatty acids, which have carboxylic acid and/or hydroxyl functionalities at both chain ends (*Graça, 2015*). When phellogen is active, the phellem cells undergo a relative quick normal sequential differential stage: expansion, massive deposition of suberin layers in the internal side of cell walls, and programmed cell death (*Soler et al., 2007*; *Teixeira & Pereira, 2010*; *Inácio et al., 2018*). Such construction and integration of suberin in cork could lead to a $^{13}$C depletion and a concomitant enrichment of $^{13}$C respired $CO_2$, strongly and directly related with temperature, during cork formation as it is well known that respiratory processes discriminate against $^{13}$C.

Another possible explanation for the $\delta^{13}$C negative correlation with temperature is that, while the tight stomatal regulation is very responsive to water stress, as in a true drought-avoiding species, meaning that trees effectively display high stomatal resistance during drought, even reducing photosynthetic metabolism which may result in the growth impairment (*Kurz-Besson et al., 2006*), the rate of carboxylation and net photosynthesis would be much less sensitive or more complacent to water stress, which would decisively influence $\delta^{13}$C of carbohydrates, and lipids.

Overall, the particularly unexpected result on $^{13}$C isotopic signature in cork might reflect the dynamics of both processes, primary $CO_2$ fixation (stomatal regulation) and the downstream metabolic processes, carbon photosynthetic assimilation or post-assimilation of carbon reserves, including the metabolites used for export and correspondent fractionation processes, at different times and under different environmental conditions. These are some alternative hypotheses which can potentially explain the varying rates of incorporation of relatively heavy carbon ($^{13}$C) in suberin during cork formation, and which merits future research.

The examination through mixed modelling of cork rings $\delta^{13}$C variations between trees (random intercept model, M2) might help to understand the range of varying rates of $^{13}$C in cork. According to our results, cork rings at the drier site (BS), in the first year after cork harvesting ($y_{cork} = 1$) were clearly different and had less negative $\delta^{13}$C ($^{13}$C enriched) than other cork-rings ($y_{cork} = 6$, at BS; or $y_{cork} = 1$ or 6, at CL) much more $^{13}$C depleted, despite sharing common variation in response to temperature (Table 2, Fig. 3). Recently

harvest trees in BS are under severe water stress due to water loss from the cork harvested stem (and branches) which much exceeds the water lost by leaf transpiration during the summer season (*Correia et al., 1992*; *Oliveira & Costa, 2012*). This suggests that under these conditions, trees would close stomata, with the consequent reduction of $CO_2$ concentration in the intercellular space, limiting $CO_2$ availability for the carboxylation, which is then forced to less discrimination and fix a relative greater proportion of the heavier isotope ($^{13}C$) (*Farquhar et al., 1989*). Also, water stress trigger suberin biosynthesis and suberization (*Aguado et al., 2012*; *Boher et al., 2018*), similarly to the promotion of xylogenesis in tree rings (*Castagneri et al., 2018*), and trees need to urgently regenerate their cork layers to survive. Under these conditions, only in the first year after cork harvest ($y_{cork} = 1$) and only at the drier site (BS), trees likely used mixing carbon pools in their metabolic pathways of suberin synthesis: from current assimilates (already relatively enriched in $^{13}C$) and from carbon reserves, which will be relatively more enriched in $^{13}C$, producing $^{13}C$ enriched cork rings.

Despite the absence of significant correlations between cork ring parameters width ($IC_{RW}$) and maximum density ($IC_{MD}$) and $\delta^{13}C$ isotopic composition, the way that trees generate cork cells or increase their walls density in response to climate conditions could be different, depending on the way they respond to water stress conditions and control stomata and/or photosynthetic activity. The $IC_{RW}$ and $IC_{MD}$ correlations with mean temperature (Table 1) were considerably weaker than the correlations found for $\delta^{13}C$ (Fig. 4). This suggests that cork is formed, or its density is increased as a result of an overall tree performance on their biomass production, while $^{13}C$ signatures reflect tree adaptive mechanisms of response to water stress and drought.

## Influence of water availability conditions on the $\delta^{18}O$ of cork rings

Under a Mediterranean type climate, with water as the main limiting factor for plant growth, distinct cork $\delta^{18}O$ should primarily reflect the variability in the water source used during growing season between trees. Furthermore, cork $\delta^{18}O$ should also be responsive to the stomatal conductance, but not sensitive to photosynthetic activity, as the $\delta^{13}C$ (*Farquhar & Lloyd, 1993*; *Roden & Farquhar, 2012*).

Cork rings $\delta^{18}O$ revealed some differences between the two study areas. At drier site (BS) cork was more depleted in $^{18}O$ than at the wetter site (CL), with higher $\delta^{18}O$ (Fig. 5). Here, these $\delta^{18}O$ higher values indicate mostly the use of a shallower, more isotopically enriched water source, but also probably some (leaf) water $^{18}O$ enrichment, given the unlimited water availability suggesting a higher stomatal conductance (Fig. 5). Moreover, the spring March-May precipitation only affected the $\delta^{18}O$ of cork rings at wetter site (CL) (Fig. 6C) suggesting some dependence on growing season precipitation for cork formation in the long and uninterrupted cork growth period (*Costa, Pereira & Oliveira, 2003*). Thus, relatively more enriched $^{18}O$ cork-rings are expected.

In contrast, at the drier site (BS), cork-rings are more $^{18}O$ depleted (Fig. 5). Moreover, no significant correlations were found with precipitation. These results might suggest that cork oak obtain their water from deeper soil horizons (*Sarris, Christodoulakis & Körner, 2007*), trees are less dependent on growing season precipitation (or even precipitation in

winter previous to growth year) for cork formation and have a strong reliance on storage carbon reserves (*Ferrio et al., 2015*; *Szymczak et al., 2019*). Other possible explanation for the non-significant correlations with precipitation is that water reserves in deeper groundwater can represent a long-term mean isotopic value of winter precipitation and hence do not only reflect the amount of winter precipitation prior to the growth season (*Szymczak et al., 2019*). Furthermore, in contrast to CL, trees at BS have a stronger stomatal conductance regulation that probably would constrain the (leaf) water $^{18}O$ enrichment.

Cork oak is a deep-rooted species, with a very effective water uptake from the seasonally recharged groundwater, and with a dual root system which allow to use of soil water, even in the drier summer season (*Vaz et al., 2010*; *David et al., 2007*). Moreover, the preferential water source of cork oak is deeper groundwater sources (*Altieri et al., 2015*). However, at BS, a direct comparison between $\delta^{18}O$ of cork rings and seasonal fluctuations of precipitation can be hampered in several ways. First, seasonal cork (suberin) synthesis is decoupled from seasonal precipitation and distinct information can be encoded in the $^{18}O$ of the cork rings; second, tree uses mixing pools of carbon which are formed with distinct water pools, for suberin synthesis in distinct growth seasons; third, depending on rooting depth, trees have to access to different water pools with varying isotopic signatures and it can be assumed that groundwater replenishment in winter previous to the growth season cannot be sufficient and trees depend also on precipitation during the growth season as other main water source.

Altogether, mixed modelling results points to the existence of a relationship between cork $\delta^{18}O$ signatures and trees preferential water sources, confirming our second hypothesis, without the complexity of the effect of climate drivers and of the cork harvesting on the variability of $\delta^{18}O$ in cork rings (Fig. 5; Table 2). However, further investigations are needed in order to clarify the interpretation of our results on the $\delta^{18}O$ signature in cork rings.

## CONCLUSIONS

Our findings on the variability of stable isotopes $\delta^{13}C$ and $\delta^{18}O$ in cork rings from (cork) harvested trees growing under contrasting water availability conditions in Mediterranean environments revealed that $\delta^{13}C$ might primarily reflect the regulation of trees' stomatal conductance and photosynthetic activity under severe water stress, whereas $\delta^{18}O$ primarily indicates the water sources used by the tree for cork growth. Cork formation is predominantly temperature-controlled, and tree uses carbon from mixing pools such as, previous year(s) synthetized and stored carbon or recent photo-assimilated carbon, in the metabolic pathways of suberin synthesis which seemed to highly discriminate $^{13}C$. Further studies would be necessary to elucidate the origin of carbon reserves, post carbonylation fractionating or carbon mobilization and the influence of environmental conditions on these processes.

The findings of this exploratory study show the potential of addressing the physiological implications of water stress for cork oak (or cork) growth through the isotope signature in cork rings, thus facilitating the evaluation of its relevance in climate change scenarios for site-specific tree adaptation.

## ACKNOWLEDGEMENTS

The authors thank Companhia das Lezírias, SA (CL, Benavente) and Herdade de Barradas da Serra (BS, Grândola) for allowing tree selection and cork sampling in their cork oak woodlands. Authors thank three anonymous reviewers and Dr. Juan Pedro Ferrio for their contribution in reviewing this manuscript.

### Funding

This work was supported by the Foundation for Science and Technology (FCT): Project IsoCork—Climate effects on cork growth assessed by isotope fingerprinting (EXPL/AGR-FOR/1220/2012); UID/AMB/04085/2013 and UIDB/00239/2020; PORLisboa2020 and Foundation for Science and Technology (FCT) (PTDC/BIA-FBT/29704-2017); FCT also supported Augusta Costa's work (SFRH/BPD/97166/2013; CEEINST/00012/2018—ENGFL). The funders had no role in study design, data collection and analysis, decision to publish, or preparation of the manuscript.

### Grant Disclosures

The following grant information was disclosed by the authors:
Foundation for Science and Technology (FCT): Project IsoCork—Climate effects on cork growth assessed by isotope fingerprinting: EXPL/AGR-FOR/1220/2012, UID/AMB/04085/2013, UIDB/00239/2020, PORLisboa2020.
Foundation for Science and Technology (FCT): PTDC/BIA-FBT/29704-2017.
Augusta Costa's work: SFRH/BPD/97166/2013, CEEINST/00012/2018—ENGFL.

### Competing Interests

The authors declare there are no competing interests.

### Author Contributions

- Augusta Costa conceived and designed the experiments, performed the experiments, analyzed the data, prepared figures and/or tables, authored or reviewed drafts of the article, and approved the final draft.
- Paolo Cherubini analyzed the data, authored or reviewed drafts of the article, and approved the final draft.
- José Graça performed the experiments, analyzed the data, authored or reviewed drafts of the article, and approved the final draft.
- Heinrich Spiecker analyzed the data, authored or reviewed drafts of the article, and approved the final draft.
- Inês Barbosa performed the experiments, prepared figures and/or tables, and approved the final draft.
- Cristina Máguas analyzed the data, authored or reviewed drafts of the article, contributed with the laboratory facilites for the isope analysis, and approved the final draft.

## Data Availability

The raw data are available in the Supplemental Files.

## Supplemental Information

Supplemental information for this article can be found online at http://dx.doi.org/10.7717/peerj.14270#supplemental-information.

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
