# Peer review of "Beyond width and density: stable carbon and oxygen isotopes in cork-rings provide insights of physiological responses to water stress in Quercus suber L"

_PeerJ, doi:10.7717/peerj.14270_

## Round 0.1 · original submission · Major Revisions

Noble Authors,

Your article was assessed by 4 independent experts. Everyone agreed that the work could be published in PeerJ, but had to be significantly improved beforehand. Please read the reviewers' comments and respond to all of them.

With best regards,

Reviewer 1 ·

Basic reporting

The manuscript shows a very interesting study on cork oak trees. The work is clearly presented, complete in all the parts. Too bad that one of the most used reference (Costa et al. 2022 Tree Physiology) is in pre-print form, therefore not available and usable for the review process.
In the text editing check in line 112: is missing ( before δ13C

Experimental design

No comment

Validity of the findings

No comment

Additional comments

The manuscript fully satisfy the three section (i.e. requirements for a good work); deserve to be published also for the importance of the topic studied and the approach (i.e. physiological approach to isotopic signature of the cork)

Reviewer 2 ·

Basic reporting

The english is clear, the introduction and background provide sufficient context, and the literature is relevant and well-referenced. The structure complies with Peerj standards and the figures are relevant and well described. Raw data is supplied.

Experimental design

The research is relevant and within the scope of the journal. The research questions are well defined, interesting and meaningful, and the research has been carried out at a high technical level. The methods are described in sufficient detail to be replicated.

Validity of the findings

The data has been provided, but the work has some statistical issues that need to be addressed. As explained later, some tests are not suitable or are incorrect: authors used univariate anovas where nested anovas are the correct choice; use of random variables with insufficient number of levels in linear mixed models. But in addition, the authors performed a lot of correlation tests with almost all combinations of climatic "windows". With only 8 trees (I understand this because isotope analysis is very expensive), the chances of type I errors are very high with multiple testing and some results don't make sense. As explained later, I suggest a hypothesis testing framework based on a more mechanistic approach, reducing the number of correlation tests in the manuscript. The manuscript will be clearer, the results less problematic and the conclusions better supported.

Major concerns
One of the main problems I have with this manuscript is the absence of a more meaningful and simpler approach to hypothesis testing. The authors performed many correlation tests with almost all combinations of climatic "windows", but with only eight trees, the chances of getting several false results due to type I errors are very high. I'll explain my point with an example, but there are several examples throughout the document. In table 1 and lines 423-425, the authors noted an " irregular " pattern due to an inverse correlation between ICRW and precipitation from July to November. Do the authors think that this correlation is due to a causal relationship? Since the precipitation in July and August is very low, and the temperatures in October and November are probably the limiting factor for photosynthesis, the biological mechanisms are not clear and the authors did not give any explanation (probably because it is a spurious result).

Also, multiple tests to get significant results make the article confusing and difficult to follow. In Figures 3 and 6, the authors show the results of the Pearson correlation tests between isotopic composition and precipitation/temperatures using climatic windows of 1, 5 and 12 months as explained in the M&M section. Most of the correlations are not significant (Figure 6 B and D). But in the text, the authors showed other climatic windows of 2, 3 or 4 months (see for example line 379 and 393-394) not shown in the figures that are significant. Avoid P-fishing because it is a dangerous road (https://doi.org/10.1038/506150a). It may be a better idea to show the results of the correlations by seasons (winter, spring, summer and autumn) and remove the results from the 5-month window to get meaningful results with a simpler and more meaningful and consistent approach.

So I think the manuscript will be clearer and much better if the hypothesis testing framework is based on a more mechanistic approach, so that only hypothesis-driven tests are shown and to avoid including every combination of climate 'windows' only to show significant results. . As mentioned, with only 8 trees, the chances of getting several type I errors are very high.

Other important statistical issues:

Authors showed the results of univariate ANOVAS on lines 284-285 and 349-350. This approach is statistically incorrect, because the samples are not independent (several samples are from the same tree). The correct approach is a nested Anova, where the identities of the trees are treated as a nesting factor. Please correct it. You will likely also get significant differences between sites, but this is the correct way.

In relation to the mixed models, the authors identified the climatic variables as fixed factors and the trees as random factors. However, you cannot identify sites as random factors because you have an insufficient number of sites. You need at least five levels (sites) to identify a variable as a random variable. With only two sites, you need to identify "site" as a fixed variable and redo all the models. This is a general rule in mixed models with random variables as described in the literature (for example, see http://doi.org/10.7717/peerj.4794).

In addition to this, I have other problems with mixed models. In lines 444-450, the authors pointed to M2 as the best model. But the best model is the M20 model. With a difference of more than 20 AIC units, M2 has very little statistical support and should be discarded. It is not clear why the authors keep the M2 model with so little support. Additionally, you can do multiple regression models with several independent climate variables (for example, variables from M20 and M3) and find the best model using a top-down approach avoiding listing several linear models with only one fixed variable (Zuur et al. (2009) Mixed Effects Models and Extensions in Ecology with R. New York: Springer).

Additional comments

Minor concerns
Introduction
Line 182: please, indicates what is the meaning of the SNIRH initials.
Results
Table 1. In the text, the results of the relationships with ICMD are shown first but are at the end of the table. I think it will be more consistent if you display it at the top of the table.
Line 342: p-value, not p-valu3
Discussion
Please, put the name of scientific names in italics.
Lines 478-491. Authors did not discuss results and only showed previous results of other works.
Lines 496-497. Why do you say that -26.8‰ is within the range of -26.3‰ and -26.1‰?
Lines 544-545. As indicated before, authors did not provide convincing evidence that M2 is statistically supported by data.
Lines 586-594. The results indicate that in the driest site (BS), the cork oaks obtain water from the deep horizons of the soil. This contradicts the initial claims (lines 174-175) and should explain more clearly why the authors think that oaks do not depend on rainfall as stated above (Costa et al. 2008; 2016).

As a general suggestion, I would like to see if the growth of cork rings is explained by the same climatic variables as "standard" tree rings in cork species. I think it is an interesting topic that is only vaguely addressed (i.e. lines 566-567), and the authors have a lot of experience with cork oak (other recent work on the relationship between cork oak growth and climate is https://doi.org/10.1007/s11104-021-05077-7).

Reviewer 3 ·

Basic reporting

The authors investigated the change in stable isotopes in cork rings from trees grown under different environmental conditions in Mediterranean environments. By analyzing the correlation between 13C or 18O and environmental parameters (e.g., temperature, precipitation and location), the authors concluded that these isotopes can be used as indicators of tree’s physiological and structural adjustments to climate changes.

This paper is well structured and clearly written. It contains relevant literature references with sufficient field background provided. The results section contain good-quality data and rational analysis, followed by detailed explanations written in the discussions section.

I think that this work could be suitable for publication in PeerJ, following minor revisions relating to suggestions below.

Experimental design

• Lines 189-190: It seems that only four trees were analyzed at each site. Could the authors comment on possible variations across the forest, and if it is statistically sufficient to analyze four trees (instead of 7+)?
• Lines 286-294: the authors pointed out that there are significant variations among the four trees selected at each site. I wonder if their corresponding results (as shown in figures 3, 4, and 5) would lead to rational analysis that is not skewed by the selection of trees (e.g., if some of the selected trees happen to the outliers among 1000+ trees in the forest)?

Validity of the findings

• lines 309-312: Is this finding species-dependent?
• Lines 337-345: Why do the correlations fluctuate between negatives and positives? Is this fluctuation equivalent to having no correlation? Is this expected or surprising? Why?

Additional comments

• Please use high-resolution images and plots.
• Across all plots, fonts used for most axis labels and marks could be bigger for easier reading.
• Figure 3 is not easy to read because of the choice of patterns (e.g., dots vs crosses vs solid blocks; then white vs grey). Please see if it would be helpful for the authors to 1) clearly mark “Grandola (BS)” on the left and “Benavente (CL)” on the right, instead of using white vs grey to distinguish between the two; and 2) use white (representing y=1) vs black (representing y=6) vs grey (representing third category) for the three types of data shown. If the authors choose to adapt this grey-scale suggestion, perhaps colored lines (e.g., in red) could be used to show the mean values.
• Figure 4A&C could be improved by labelling “previous growth year” on the left, and “current growth year” on the right directly on these sub figures.

·

Basic reporting

Language and general structure of the MS is of good quality, including clear figures and the relevant raw data. Main weakness is that the discussion shows some inconsistencies.

1.1) My main concern is the interpretation of cork isotope signal as directly comparable with tree rings (e.g. line 496-497). Since cork derives from lipids, it is expected to be more depleted than carbohydrate-derived wood cellulose. I also recommend to revise and expand the discussion on the potential effect of post-photosynthetic fractionation in cork (e.g. lines 507-514). Some key information is needed to interpret this, e.g. what are the storage compounds used in cork formation? are they derived from starch reserves, or stored in another form? This is relevant, e.g. in lines 523-525, where the depletion of cork seems to be attributed to the season of formation, but is most likely due to its metabolic origin. Indeed, stored carbon in the form of starch (used in earlywood) is usually enriched in 13C, rather than depleted, as stated in the MS (see e.g. Helle and Schleser 2004; Eglin et al. 2010; Offermann et al. 2011; further refs. recently reviewed in Gessler & Ferrio 2022).

1.2) In general, the results comparing climate variables and isotope composition could benefit from a deeper insight into their physiological meaning. For example, in lines 586-587 it is not clear why we should expect a correlation between d18O and precipitation (rather than the d18O of precipitation). Besides, the interpretation of the eventual role of in-leaf evaporative enrichment (581-582) is not clear: higher transpiration (due to more open stomata) does not generally imply higher evaporative enrichment, and indeed, due to the "Péclet effect", the opposite could be the case, due to a greater input of non-enriched water into the leaf.

Refs cited:

Eglin T, Francois C, Michelot A, Delpierre N, Damesin C (2010) Linking intra-seasonal variations
in climate and tree-ring δ13C: a functional modelling approach. Ecol Model 221:1779–1797

Gessler, A., Ferrio, J.P. (2022). Postphotosynthetic Fractionation in Leaves, Phloem and Stem. In: Siegwolf, R.T.W., Brooks, J.R., Roden, J., Saurer, M. (eds) Stable Isotopes in Tree Rings. Tree Physiology, vol 8. Springer, Cham. https://doi.org/10.1007/978-3-030-92698-4_13

Helle G, Schleser GH (2004) Beyond CO2-fixation by Rubisco—an interpretation of13C/12C variations
in tree rings from novel intra-seasonal studies on broad-leaf trees. Plant Cell Environ
27:367–380

Offermann C, Ferrio JP, Holst J, Grote R, Siegwolf R, Kayler Z, Gessler A (2011) The long
way down—are carbon and oxygen isotope signals in the tree ring uncoupled from canopy
physiological processes? Tree Physiol 31:1088–1102

Experimental design

Experimental design is sound, and based on a clear research question. Regarding the methodology, I just have one minor concern:

2.1) Please clarify how the cork-rings were dated. Annual rings are visible, but (apparently) used only to date within each cork growth (harvesting) period. However, it is not clear how this information (harvest years) was obtained. Was it recorded by the forest managers, or defined a posteriori from the anatomical changes in the cork?

Validity of the findings

No comment.

Additional comments

No comment.

---

## Round 0.2 · accepted · Accept

Dear Authors,

I am pleased to inform you that all reviewers agreed - your work may be published in PeerJ. Congratulations!

Reviewer 2 ·

Basic reporting

The English is clear, the introduction and background provide sufficient context, and the literature is relevant and well-referenced. The structure complies with Peerj standards and the figures are relevant and well described. Raw data is supplied.

Experimental design

The research is relevant and within the scope of the journal. The research questions are well defined, interesting and meaningful, and the research has been carried out at a high technical level. The methods are described in sufficient detail to be replicated.

Validity of the findings

The data has been provided, and the statistical issues has been addressed.

Reviewer 3 ·

Basic reporting

The authors have addressed all reviewers' comments.

Experimental design

The authors have addressed all reviewers' comments.

Validity of the findings

The authors have addressed all reviewers' comments.

·

Basic reporting

In this revised version, my main concerns on the manuscript, mostly related to data interpretation, have been solved, including a deeper discussion on the mechanisms behind observed isotopic variability in cork. In particular, the role of postphotosynthetic fractionation and leaf isotopic enrichment has been adequately addressed, with a level of detail that is realistic for the temporal scale and the physiological information available.


1.2) In general, the results comparing climate variables and isotope composition could benefit from a deeper insight into their physiological meaning. For example, in lines 586-587 it is not clear why we should expect a correlation between d18O and precipitation (rather than the d18O of precipitation). Besides, the interpretation of the eventual role of in-leaf evaporative enrichment (581-582) is not clear: higher transpiration (due to more open stomata) does not generally imply higher evaporative enrichment, and indeed, due to the "Péclet effect", the opposite could be the case, due to a greater input of non-enriched water into the leaf.

Experimental design

In general the methods were already well described in the original version. The additional details requested on the dating of cork-rings have been provided, clarifying this issue.

Validity of the findings

No comment

Additional comments

No comment